# A first calibration of the JULES-crop version 7.4 for rice

# 2 using the novel O<sub>3</sub>-FACE experiment in China

- Beiyao Xu<sup>1,2</sup>, Steven Dobbie<sup>2</sup>, Huiyi Yang<sup>3,4</sup>, Lianxin Yang<sup>5</sup>, Yu Jiang<sup>6</sup>, Andrew Challinor<sup>2</sup>, Karina Williams<sup>4,7</sup>,
- Yunxia Wang<sup>8</sup>, Tijian Wang<sup>1</sup>
- 1 School of Atmospheric Sciences, Nanjing University, China
- 2 Institute for Climate and Atmospheric Science, University of Leeds, UK
- 3 Natural Resources Institute, University of Greenwich, UK
- 4 Global Systems Institute, University of Exeter, UK
- 5 Key Lab of Crop Genetics & Physiology of Jiangsu Province, Yangzhou University, China
- 6 Jiangsu Collaborative Innovation Center for Modern Crop Production, Nanjing Agricultural University, China
- 7 Met Office, Exeter, UK
- 8 College of Environmental Science and Engineering, Yangzhou University, China
- Correspondence to: Lianxin Yang (lxyang@yzu.edu.cn) and Steven Dobbie (J.S.E.Dobbie@leeds.ac.uk)
- Abstract. Ozone (O<sub>3</sub>) pollution poses an escalating threat to rice production and food security in China, with
- concentrations projected to rise under future climate scenarios. Accurately quantifying O<sub>3</sub> impacts on rice is thus
- crucial for informed agricultural planning. This study is the first to utilise Free Air Concentration Enrichment
- (FACE) observations specific to rice for calibrating a crop model (JULES-crop) and assessing the impacts of O<sub>3</sub>.
- FACE experiments, which involve growing crops under natural field conditions while exposing them to elevated
- O<sub>3</sub> levels, provide an ideal approach for studying the effects of O<sub>3</sub> on crops. Utilising data from the only O<sub>3</sub>-FACE
- facility dedicated to rice, we calibrated physiological and O<sub>3</sub>-response parameters in JULES-crop and evaluated
- the model against additional independent FACE observations. The calibration establishes this as the first crop
- model refined with ideal open-air field observations, significantly enhancing its capability to simulate rice growth
- processes and O<sub>3</sub>-induced yield losses, surpassing the performance of simulations based on the default parameters
- in JULES-crop. With this newly calibrated model, JULES-crop is now equipped to assess the impacts of O<sub>3</sub> on
- agriculture, offering a valuable tool to inform mitigation strategies.

#### 1 Introduction

- Rice is the staple food for over half of the world's population and plays a crucial role in global food security. The
- rising concentration of ozone (O<sub>3</sub>) is a major concern, contributing to significant losses in crop production
- worldwide (Van Dingenen et al., 2009). Mills et al. (2018) estimated that the average global yield loss of rice due
- to O₃ was 4.4% between 2010 and 2012. In China, O₃ caused relative rice yield losses of 6.2–52.9% between 2014
- and 2018, and 23% between 2017 and 2019 (Feng et al., 2022; Xu et al., 2021). Consequently, assessing the impact
- of O<sub>3</sub> on rice growth is essential, especially as O<sub>3</sub>-polluted areas overlap with crop-growing regions and pose a
- long-term threat to food security (Emberson et al., 2018).
- The main O<sub>3</sub> dose-response functions used to assess rice yield loss include concentration-based methods, such as
- the accumulated dose of O<sub>3</sub> over 40 ppb (AOT40) and the daily mean seven-hour concentrations (M7), and flux-
- based methods, such as the phytotoxic O<sub>3</sub> dose (POD) (Tai et al., 2021). Both concentration-based and flux-based
- methods can establish a relationship with relative yield loss based on field experiments. The relationship between
- relative yield loss and O<sub>3</sub> level, known as the O<sub>3</sub> response function, is a valuable tool that underpins extensive
- research into crop yield losses caused by O<sub>3</sub> exposure (Ramya et al., 2023).
- Some crop models have incorporated O<sub>3</sub> parameters to better understand its impacts (Guarin et al., 2024; Leung
- et al., 2020; Ewert and Porter, 2000). For instance, the Decision Support System for Agrotechnology Transfer
- (DSSAT) crop model established an O<sub>3</sub> stress factor using the M7 metric (Guarin et al., 2024). GLAM-ROC
- simulated O<sub>3</sub> effects by reducing evapotranspiration, transpiration efficiency, and harvest index based on AOT40
- metric (Droutsas et al., 2020). The Joint UK Land Environment Simulator with crops (JULES-crop) integrated a
- 20 flux-based O<sub>3</sub> damage scheme developed by Sitch et al. (2007) to assess reductions in net photosynthesis. Flux-
- 21 based methods account for stomatal conductance and environmental conditions, such as temperature and vapour
- pressure deficit, to modify O<sub>3</sub> uptake and thus directly link absorbed O<sub>3</sub> dose to physiological damage. Compared
- with concentration-based methods, flux-based methods exhibit enhanced performance in correlating O<sub>3</sub> levels
- with relative yield loss, enabling more precise assessments (Pleijel et al., 2004; Pleijel et al., 2022; Mills et al.,
- 2011; Ronan et al., 2020). Nonetheless, O<sub>3</sub>-related parameters in crop models require calibration to ensure reliable
- performance, even when using a flux-based O<sub>3</sub> scheme.
- Open-top chambers (OTC) and free air concentration enrichment (FACE) experiments are two major methods
- used to help calibrate parameters in crop models. State-of-the-art FACE experiments, which provide more natural
- environments for crops, are ideal for establishing O<sub>3</sub> exposure metrics and investigating the impacts of O<sub>3</sub> on crops
- (Montes et al., 2022; Feng et al., 2018). To date, only four O<sub>3</sub>-FACE facilities have been established for crops
- worldwide (Montes et al., 2022): wheat and rice experiments in China (Tang et al., 2011), wheat experiments in
- India (Yadav et al., 2019), grape experiments in Italy (Moura et al., 2023), and soybean experiments in the United
- States (Aspray et al., 2023). However, the rice-specific O<sub>3</sub>-FACE experiment has not yet been used to calibrate
- any crop models.
- The parameterisation of crops in JULES was developed by Osborne et al. (2015). JULES-crop incorporates flux-
- based O<sub>3</sub> exposure metrics to analyse the loss of accumulated carbon based on the exact O<sub>3</sub> flux entering the crop
- stomata, which is influenced by environmental conditions (Sitch et al., 2007). The impact of O<sub>3</sub> on crops is also
- reflected in reductions in crop height, leaf area index (LAI), and crop yields. Additionally, Tai et al. (2021)

- highlighted that mechanistic crop models such as JULES-crop can combine the fertilisation effects of atmospheric
- carbon dioxide (CO<sub>2</sub>) with the O<sub>3</sub> influence. Thus, JULES-crop is a suitable tool for investigating the effects of
- O<sub>3</sub> on crops, accounting for environmental factors that modify the mechanisms of O<sub>3</sub> effects (Leung et al., 2022).
- However, the crop growth and development parameters for rice, as well as the O<sub>3</sub> impact parameters within
- JULES-crop, have not yet been calibrated. Calibrating JULES-crop would enhance its performance in simulating
- rice production under O<sub>3</sub> influence.
- In this research, we calibrated the rice parameters in JULES-crop using novel O<sub>3</sub>-FACE data, enabling leading-
- edge future assessments of O<sub>3</sub> damage to rice. The study has three key objectives: (1) to calibrate JULES-crop
- using novel O<sub>3</sub>-FACE field data; (2) to evaluate the model's performance in capturing crop growth characteristics
- using independent observations; and (3) to assess the impact of O<sub>3</sub> on rice physiology, phenology, and yields. This
- research enhances understanding of the mechanisms through which O<sub>3</sub> affects rice growth and development,
- providing a stronger basis for characterising the future impact of O<sub>3</sub> on rice production.

#### 2 Method

# 2.1 Description of the JULES-crop

- JULES-crop is an extension of JULES, a land surface model designed to simulate the fluxes of carbon, water,
- energy, and momentum between the land surface and the atmosphere (Best et al., 2011; Clark et al., 2011). JULES-
- crop was developed to simulate the growth and development of major crops, including wheat, soybean, maize,
- and rice, under a range of environmental influences such as temperature, precipitation, radiation, and soil moisture
- (Osborne et al., 2015). Its structure, illustrated in Fig. 1, incorporates the physiological processes of crops,
- including photosynthesis, respiration, and biomass accumulation.
- JULES-crop simulates the physiological and phenological processes of crops, predicting yields at both field and
- global scales. This capability makes it a valuable tool for understanding the impacts of climate change and air
- pollution on agriculture (Leung et al., 2022; Wolffe et al., 2021; Vianna et al., 2022). To date, winter wheat (in
- preparation), maize (Williams et al., 2017), and soybean (Leung et al., 2020) within JULES-crop have been
- calibrated using observational data. Mathison et al. (2021) updated several rice and wheat parameters in JULES-
- crop, relying primarily on literature, but did not account for O<sub>3</sub> effects. In this study, novel O<sub>3</sub>-FACE experimental
- data was utilised to calibrate rice parameters in JULES-crop for the first time, improving its ability to assess O<sub>3</sub>
- impacts on rice growth.

Figure 1. Schematic of JULES-crop.

1

- 3 JULES-crop utilises a flux-based approach to simulate the O<sub>3</sub> damage following Sitch et al. (2007). It assumes
- 4 that the potential net photosynthesis  $A_p$  was suppressed by O<sub>3</sub>:

$$A = A_p F$$

where A is leaf-level net photosynthesis with the  $O_3$  effects and F is the reduction factor:

$$F = 1 - a \cdot max [F_{O_3} - F_{O_3crit}, 0]$$

- 8 where  $F_{0_3}$  represents instantaneous leaf uptake of O<sub>3</sub>.  $F_{0_3crit}$  and a are plant-functional-type-specific threshold
- 9 and sensitivity factor respectively.
- 10 The O<sub>3</sub> flux  $F_{O_3}$  (nmol m<sup>-2</sup> s<sup>-1</sup>) is calculated as:

$$F_{O_3} = \frac{[O_3]}{r_a + \left[\frac{\kappa_{O_3}}{q_I}\right]}$$

- where  $[O_3]$  (nmol m<sup>-3</sup>) is the molar O<sub>3</sub> concentration at the reference level,  $r_a$  (s m<sup>-1</sup>) is the aerodynamic
- 13 resistance and the boundary layer resistance between the leaf surface and reference level (Monin and Obukhov,
- 14 1954),  $\kappa_{03}$  is the ratio of leaf conductance for O<sub>3</sub> to leaf conductance for water vapour (1.67), and  $g_l$  represents
- 15 the leaf conductance for H<sub>2</sub>O as a linear function of photosynthetic rate (Cox et al., 1999):

$$g_l = g_l^* F$$

where  $g_l^*$  is the leaf conductance in the absence of O<sub>3</sub> effects.

# 2.2 O<sub>3</sub>-FACE experiments

- 19 The O₃-FACE experiment was conducted in Xiaoji, China (32°35'5"N, 119°42'0"E) in 2012. It features four
- 20 regular octagonal O<sub>3</sub>-FACE fields (14 m in diameter) and four control fields, each covering an area of
- 21 approximately 120 m². The experimental fields are spaced over 70 m apart to minimise the influence of O₃ release
- on neighbouring fields. Pipes positioned 50–60 cm above the crops released pure O<sub>3</sub> gas into each O<sub>3</sub>-FACE field
- 23 between 09:00 and 16:00 during the rice growing period. The mean daytime O₃ concentration during the

- experimental period was approximately 46 ppb under the elevated O<sub>3</sub> treatment, compared to 37 ppb in the ambient
- environment—an increase of around 25%. The environmental conditions in the O<sub>3</sub>-FACE and control fields were
- identical, except for the presence of O<sub>3</sub> pipes in the O<sub>3</sub>-FACE fields. Samples from the O<sub>3</sub>-FACE fields were
- collected from the field centre, at least 1.5 m away from the O<sub>3</sub> pipes, to ensure that the sampled rice had grown
- under stable O<sub>3</sub> conditions. Further details of the O<sub>3</sub>-FACE system can be found in Wang et al. (2012).
- The rice cultivar used was II You 084. The rice was planted on 30th May 2012 and reached maturity on 19th
- October 2012 in the ambient O<sub>3</sub> environment and 12th October 2012 in the elevated O<sub>3</sub> environment. During the
- growth period, key developmental stages, such as jointing and flowering, were recorded, and crop growth
- characteristics—including dry biomass of leaves, stems, and panicles, leaf area index, and plant height—were
- measured at these stages to calibrate the model.
- Three planting densities were employed during transplantation: low density (16 plants m<sup>-2</sup>), medium density (24
- plants m<sup>-2</sup>), and high density (32 plants m<sup>-2</sup>). In addition to standard growth measurements, photosynthesis-related
- variables—including leaf temperature, internal leaf CO<sub>2</sub> concentration, stomatal conductance, and photosynthesis
- rate (CO<sub>2</sub> assimilation rate)—were assessed using a LI-6400 portable photosynthesis system. After the rice
- 15 reached maturity, 64 plants from each experimental field were harvested and dried to calculate the average rice
- 16 yields.

17

25

# 2.3 FACE experiment for JULES-crop evaluation

- Following calibration, observations of rice yields, height, and the dry weight of leaves, stems, and panicles from
- an independent FACE experiment were then used to evaluate the performance of JULES-crop. These additional
- 20 field experiments were conducted in Danyang, China (31°54′31″N, 119°28′21″E), and provided rice data for the
- 2022 and 2023 growing seasons. Two cultivars, Yangdao 6 and Wuyungeng 23, were transplanted on 20th July
- 2022 and 21st July 2023, respectively, and harvested between late October and early November. Yangdao 6 is an
- Indica rice cultivar, while Wuyungeng 23 belongs to the Japonica subspecies group, both of which represent the
- two major rice subspecies cultivated in China.

# 2.4 Data preparation

- JULES-crop requires driving data, ancillary data, and control files to configure the model. Observations of hourly
- air pressure, specific humidity, air temperature, precipitation, wind speed, and shortwave radiation (SW) recorded
- during the O<sub>3</sub>-FACE experiments were used as driving data. Diffuse radiation was calculated using a constant
- diffuse fraction in the model, with the default value of 0.4 applied in this study due to the absence of observational
- data. Surface downward longwave radiation (LW) was not measured in the O<sub>3</sub>-FACE experiment and was instead
- estimated using an empirical model based on local observations (Chang and Zhang, 2019):

32 
$$R_{\downarrow} = \sigma \cdot (T_a)^4 \cdot \left[ clf + (1 - clf) \cdot \left( a \cdot \ln \left( \frac{e_a}{T_a} \right) + b \cdot \varphi + c \right) \right]$$

- where  $R_{\downarrow}$  is the downward LW under all kinds of sky (clear and cloudy),  $T_a$  is the air temperature;  $e_a$  is the water
- vapour pressure;  $\varphi$  is the relative humidity;  $\sigma$  is the Stefan-Boltzmann constant; a, b, and c are the empirical
- 35 coefficients (Table 1); and *clf* is the cloud modification factor, set to 0 under clear sky conditions:

$$clf = 1 - K_t$$

where  $K_t$  is the clearness index which was calculated as follows:

$$K_t = \frac{H_m}{H_0}$$

- 4 where  $H_m$  represents the hourly measured solar radiation, and  $H_0$  denotes the hourly extraterrestrial solar radiation.
- 5 Detailed calculation for  $H_0$  can be found in Kumar and Umanand (2005).

#### Table 1 Empirical coefficients used in the longwave radiation model.

| Period                        | a     | b      | С     |
|-------------------------------|-------|--------|-------|
| Daytime with the cloud impact | 0.118 | 0      | 1.033 |
| Nighttime                     | 0.08  | 0.0014 | 1.026 |

7

8

9

10

6

For the ancillary data, soil property values were extracted from the ancillary dataset used in the HadGEM2-ES model, which also underpins global simulations (Osborne et al., 2015). Another crucial factor influencing crop growth, the annual average CO<sub>2</sub> concentration, was set based on data provided by the Global Monitoring

- Laboratory (GML) of the National Oceanic and Atmospheric Administration (NOAA).
- The weather station for the evaluation experiments provided only daily temperature and precipitation data.
- Consequently, additional meteorological variables, including wind, humidity, and longwave radiation, were
- sourced from the ECMWF Reanalysis v5 (ERA5) dataset. However, the ERA5-generated shortwave radiation
- (SW) for 2022 and 2023 disrupted the JULES-crop simulations leading to unrealistically high leaf area index
- (LAI) values (exceeding 15). The overestimation of SW in ERA5 has been widely reported, with studies
- attributing it to the omission of aerosol variations and a limited capacity to simulate clouds and water vapour,
- resulting in an overestimation of hourly SW in China by approximately 73.95 W m<sup>-2</sup> (He et al., 2021; Jiang et al.,
- 2020; Tong et al., 2023; Li et al., 2023). To address this, SW was bias-corrected using observations from the O<sub>3</sub>-
- FACE experiment conducted in 2012.
- Additionally, O<sub>3</sub> concentration observations were unavailable for the evaluation experiments. Hourly O<sub>3</sub> data from
- the nearest station of the China National Environmental Monitoring Centre (https://www.cnemc.cn/) were used
- instead. Aside from these driving data, e.g. weather variables, O<sub>3</sub> concentrations, CO<sub>2</sub> concentrations, and crop
- stage dates, the evaluation simulations applied the same settings and parameters as those used in the calibration.

## 3 Results

#### 3.1 Calibration

- All parameters calibrated using the O<sub>3</sub>-FACE experiment are listed in Tables 2 and 3. The calibration process for
- rice involved four main steps. First, leaf-level simulations were calibrated by fitting simulated photosynthesis
- rates with observed values. Nitrogen content in leaves, stems, and roots was obtained from observations and
- literature. Observed leaf temperature, internal CO<sub>2</sub> concentration, and stomatal conductance were used as model

- inputs. Photosynthesis-related parameters were adjusted based on discrepancies between observed and simulated
- photosynthesis rates. Notably, O<sub>3</sub> damage was not considered during this step.
- Second, canopy-level simulations were calibrated by determining the rice growth rate and partitioning of
- assimilated carbon. Air temperature data were used to calculate the accumulated temperature required for rice
- growth stages and the allocation of carbon to various carbon pools was also defined during this phase.
- Third, model simulations were evaluated against observed LAI and crop height following the calibration of crop
- physiology parameters. Lastly, rice yields were compared with observations under both ambient and elevated O<sub>3</sub>
- concentrations.

The calibration process involved iteratively adjusting parameters manually until the model simulations fell within the range of observed values. Additional adjustments were made to refine results, aiming to align them closer to the central tendency of the observations. Although the number of simulations was constrained by computational limitations, the process successfully achieved agreement with all available observations, ensuring no discrepancies remained. While finer and finer incremental adjustments were not feasible due to computational limitations, the

14 approach effectively balanced precision and generalisation, capturing the essential crop observations without

15 overfitting.

16

17

Table 2 Calibrated plant functional types (PFT) parameters representing rice.

| Parameters    |                | Osborne et al. | This study  | The meening of peremeters                                  |  |  |
|---------------|----------------|----------------|-------------|------------------------------------------------------------|--|--|
| Parameters    |                | (2015)         | Tills study | The meaning of parameters                                  |  |  |
| $n_l$         | nl0_io         | 0.073          | 0.065       | Leaf nitrogen concentration (kg N/kg C).                   |  |  |
| "             | ns nl io       | 1              | 0.52        | Ratio of stem nitrogen concentration to leaf               |  |  |
| $\mu_{sl}$    | lis_iii_io     | 1              |             | nitrogen concentration.                                    |  |  |
| ,,            | nr_nl_io 1     | 1              | 0.46        | Ratio of root nitrogen concentration to leaf               |  |  |
| $\mu_{rl}$    |                | 1              |             | nitrogen concentration.                                    |  |  |
| 20            | <b>c</b> :- 0E | 8E-4           | 1.28E-3     | Scale factor relating $V_{\text{cmax}}$ with leaf nitrogen |  |  |
| $n_e$         | neff_io        | 6E-4           | 1.20E-3     | concentration.                                             |  |  |
| $f_{dr}$      | fd_io          | 0.015          | 0.008       | Scale factor for dark respiration.                         |  |  |
| T             | tuma is        | 26             | 20          | Upper temperature parameter for                            |  |  |
| $T_{upp}$     | tupp_io        | 36             | 38          | photosynthesis (°C).                                       |  |  |
| $q_{10,leaf}$ | q10_leaf_io    | 2              | 2.1         | Q10 factor for plant respiration.                          |  |  |

# Table 3 Calibrated crop-related parameters representing rice.

| Parameters      | Osborne et        |         | This study | The meaning of parameters                                                                                                                     |  |
|-----------------|-------------------|---------|------------|-----------------------------------------------------------------------------------------------------------------------------------------------|--|
|                 |                   | (2015)  |            |                                                                                                                                               |  |
| $TT_{emr}$      | tt_emr_io         | 60      | 50         | Thermal time between sowing and emergence (°C d).                                                                                             |  |
| $TT_{veg}$      | tt_veg            | 980*    | 1300       | Thermal time between emergence and flowering (°C d).                                                                                          |  |
| $TT_{rep}$      | tt_rep            | 653*    | 880        | Thermal time between flowering and harvest (°C d).                                                                                            |  |
| $lpha_{root}$   | alpha1_io         | 18.5    | 17.4       | ,                                                                                                                                             |  |
| $lpha_{stem}$   | alpha2_io         | 19.0    | 17.4       |                                                                                                                                               |  |
| $\alpha_{leaf}$ | alpha3_io         | 19.5    | 17.9       | Coefficient for determining partitioning.                                                                                                     |  |
| $eta_{root}$    | beta1_io          | -19.0   | -20        |                                                                                                                                               |  |
| $eta_{stem}$    | beta2_io          | -17.0   | -16.7      |                                                                                                                                               |  |
| $eta_{leaf}$    | beta3_io          | -18.5   | -18.5      |                                                                                                                                               |  |
| γ               | gamma_io          | 20.9    | 24.5       | Coefficient for determining specific lea                                                                                                      |  |
| δ               | delta_io          | -0.2724 | -0.145     | area (m²/kg).                                                                                                                                 |  |
| τ               | remob_io          | 0.25    | 0.12       | Remobilisation factor. Fraction of ster                                                                                                       |  |
|                 |                   |         |            | growth partitioned to reserve carbon.                                                                                                         |  |
| $f_{C,stem}$    | cfrac_s_io        | 0.5     | 0.404      | Carbon fraction of dry matter for stems.                                                                                                      |  |
| $f_{C,root}$    | cfrac_r_io        | 0.5     | 0.337      | Carbon fraction of dry matter for roots.                                                                                                      |  |
| $f_{C,leaf}$    | cfrac_l_io        | 0.5     | 0.399      | Carbon fraction of dry matter for leaves.                                                                                                     |  |
| κ               | allo1_io          | 1.4     | 1.27       | Allometric coefficient relating stem carbon                                                                                                   |  |
| λ               | allo2_io          | 0.4     | 0.24       | to crop height.                                                                                                                               |  |
| μ               | mu_io             | 0.05    | 2          | Allometric coefficient for calculation of senescence.                                                                                         |  |
| ν               | nu_io             | 0       | 6          | Allometric coefficient for calculation of senescence.                                                                                         |  |
| $f_{yield}$     | yield_frac_io     | 1.0     | 0.8        | Fraction of the harvest carbon pool converted to yield carbon (yield is the economically valuable component of the harvest pool e.g. kernel). |  |
| $C_{init}$      | initial_carbon_io | 0.01    | 0.01       | Carbon in crop at emergence in kgC/m <sup>2</sup> .                                                                                           |  |
| $DVI_{init}$    | initial_c_dvi_io  | 0.0     | 0.1        | Development Index (DVI) at which the crop carbon is set to initial carbon io.                                                                 |  |
| $DVI_{sen}$     | sen_dvi_io        | 1.5     | 1.25       | DVI at which leaf senescence begins.                                                                                                          |  |

<sup>2 \*</sup> These parameters were spatially varying in Osborne et al. (2015).

### 3.1.1 Photosynthesis

- The potential leaf-level photosynthesis, unaffected by water stress and O<sub>3</sub> effects, is calculated based on three
- potentially limiting rates: Rubisco-limited rate  $(W_c)$ , light-limited rate  $(W_l)$ , and the rate of transport of
- photosynthetic products  $(W_e)$  for  $C_3$  plants, as detailed in Clark et al. (2011).
- Following Farquhar et al. (1980) and Collatz et al. (1991), several parameters in the photosynthesis scheme are
- temperature-dependent, including the maximum rate of Rubisco carboxylation,  $V_m$  (mol  $CO_2$   $m^{-2}$  s), which is
- critical for calculating both  $W_c$  and  $W_e$ .  $V_{cmax}$  is calculated assuming an optimal temperature range defined by
- $T_{upp}$  and  $T_{low}$ .

$$V_{cmax} = \frac{V_{cmax25} f_T(T_c)}{\left[1 + e^{\{0.3(T_c - T_{upp})\}}\right] \left[1 + e^{\{0.3(T_{low} - T_c)\}}\right]}$$

- where  $V_{cmax25}$  represents the maximum rate of carboxylation of the enzyme Rubisco at 25°C and is assumed to
- be linear dependent on the leaf nitrogen concentration. For  $C_3$  crop,  $V_{cmax25} = n_e n_l$ , where  $n_e$  is the scale factor
- and  $n_l$  is the leaf nitrogen concentration (kg N/kg C).  $T_c$  is the leaf temperature in °C,  $T_{upp}$  and  $T_{low}$  are PFT-
- dependent parameters, and  $f_T$  depends on the parameter  $q_{10,leaf}$ , the factor by which plant respiration increases
- by a 10°C increase in temperature:

$$f_T = q_{10,leaf}^{0.1(T_c - 25)}$$

- 16 Changes in PFT parameters primarily influences the simulations of photosynthesis rate, which in turn affects the
- 17 accumulation of carbon in rice. In JULES-crop, the photosynthesis process is closely linked to the nitrogen content
- 18 of the crop. Leaf nitrogen concentration  $(n_l)$  is a key factor impacting the photosynthesis rate and was estimated
- based on literature sources (Fig.2a). As leaf nitrogen concentration declines from the vegetative to the ripening
- stage, the rice plant's capacity for carbon accumulation diminishes.
- The ratio of the nitrogen content of roots relative to leaves  $(\mu_{rl})$  was also derived from literature (Fig.2b). This
- 22 ratio determines the nitrogen content in the roots, which further influences the respiration rate. The maturity stage
- 23 was excluded when calculating the average values for each stage. The values presented in Fig.2 were collected
- 24 from peer-reviewed studies conducted across China over the past 20 years (listed in the supplementary file),
- 25 encompassing several rice cultivars grown in major rice-producing regions.

Figure 2. Leaf nitrogen concentration (kg N/kg C) (left) and the ratio of root nitrogen concentration to leaf nitrogen concentration (right). The grey dashed line represents the values selected for the simulation, while the dots indicate the observed values.

The ratio of the nitrogen content of stems to leaves ( $\mu_{sl}$ ) was determined from the O<sub>3</sub>-FACE observations. The ratio varied across growth stages, reaching its highest value during the maturity stage (Fig. 3). This is because at maturity the leaves consist solely of yellow leaves, which have lower nitrogen content compared to the green leaves present during earlier stages. The calibrated  $\mu_{sl}$  is the average value during the tillering, jointing, and heading stages.

**Figure 3.** Ratio of the stem nitrogen concentration to the leaf nitrogen concentration. The grey dashed line and the dots show the values for the simulation and observations respectively.

The simulations of net leaf photosynthesis rate, using the default parameters from Osborne et al. (2015), underestimated the observed values (Fig. 4a). Several parameters including  $n_l$ ,  $n_e$ ,  $f_{dr}$ ,  $T_{upp}$ , and  $q_{10,leaf}$ , were calibrated to make the simulation results in better agreement with observations. The standard photosynthesis model assumes that the upper temperature limit for C3 crops is 36°C. However, when the temperature exceeded

36°C, the simulated photosynthesis rates were still underestimated (Fig. 4b). This suggests that temperatures above 36°C should be increased to 38°C to obtain improved agreement with observations, as shown in Figures 4c and 5.

**Figure 4.** Simulated photosynthesis rate ( $\mu mol\ CO_2\ m^{-2}\ s^{-1}$ ) using parameters before (a) calibration and after calibration without (b) or with (c) changing the upper temperature limitation parameter ( $T_{upp}$ ). The dashed line is the 1:1 line.

Figure 5 shows that the simulated leaf photosynthetic rate starts to decrease at approximately 30 °C using the calibrated temperature parameters while the simulated curves using the default  $T_{upp}$  from Osborne et al. (2015) reached the optimum temperature at about 29 °C. The exact optimum temperature for simulations varied with the intercellular CO<sub>2</sub> concentration of leaves (Ci). According to the experimental data collected from the literature, the optimum temperature should be around 30 °C, depending on the environmental conditions such as nitrogen content of leaves, light intensity, and CO<sub>2</sub> concentration as well as growth stages. After calibration, the response of leaf photosynthetic rate to leaf temperature was closer to observations both from this study and the literature.

- 1 Figure 5. The coloured lines are simulated temperature responses of photosynthesis rate using the mean value of the
- observed intercellular CO<sub>2</sub> concentration of leaves (Ci) and calibrated (38 °C) or default (36 °C)  $T_{upp}$ . The filled dots and
- open circles represent the observations used in this study and simulations generated by calibrated parameters, respectively.
- The error bars were taken from five independent studies (Table S1), which span multiple rice cultivars, nitrogen regimes,
- CO<sub>2</sub> levels, and light intensity.

# 3.1.2 Rice development and assimilate partitioning

- The development status of rice is closely linked to its phenological progression and is represented by the
- Development Index (DVI). The DVI increases as the ratio of accumulated thermal time to the prescribed thermal
- time for each developmental phase rises. Initially, the DVI is set to −1 at sowing, increases to 0 at emergence,
- completes accumulation before flowering at a value of 1, and reaches a value of 2 at maturity.
- Once rice is sown, its developmental rate, defined by the DVI, depends on the prescribed thermal time, which
- includes the thermal time between sowing, emergence, flowering, and maturity stages (Osborne et al., 2015). The
- 13 thermal time  $(T_{eff})$  can be calculated as follows:

$$T_{eff} = \begin{cases} 0 & for \ T 

Figure 6. Fraction of daily accumulated net primary productivity partitioned to roots (purple), stems (blue), leaves (yellow),

- and harvested parts (red) of the crop as a function of development index (DVI; 0 = emergence, 1 = flowering, 2 = maturity)
- for rice. The black dashed line is the fraction based on parameters used in Osborne et al. (2015).

The accumulated carbon in different carbon pools directly affects the biomass of various rice organs. The model

- calculates carbon accumulation and distribution, so the fractions of carbon-to-dry matter in the root, stem, and
- leaf  $(f_{C,root}, f_{C,stem}, \text{ and } f_{C,leaf})$  must be defined prior to running the model. The values used in our calibrated
- simulations were taken from the observations and are listed in Table 2, along with the default values from Osborne
- et al. (2015). The value of the carbon fraction impacts the root growth, crop height, and LAI.

# 3.1.3 LAI and crop height

1

- 11 Leaf Area Index (LAI) is an important attribute of crops, reflecting their capacity for carbon accumulation. In
- 12 JULES-crop, LAI is linked to the leaf carbon pool (Osborne et al., 2015):

$$LAI = \frac{C_{leaf}}{f_{c,leaf}} SLA$$

- where  $C_{leaf}$  indicates the amount of carbon in leaves,  $f_{c,leaf}$  represents the carbon fraction of dry matter in leaves,
- and SLA is the specific leaf area (m<sup>-2</sup> leaf kg<sup>-1</sup>):

$$SLA = \gamma (DVI + 0.06)^{\delta}$$

- where  $\gamma$  and  $\delta$  were determined by fitting the curve between DVI and SLA (De Vries et al., 1989) from
- 18 observations (Fig.7).

2 Figure 7. Specific leaf area against development index. Coloured symbols indicate observations, and the colour shows the

data from different experiment fields. The black dashed line and the black solid line show the fit using parameters from

Osborne et al. (2015) and our calibrated parameters, respectively. Note that various symbols correspond to successive

5 sampling dates from the same experimental field, thereby illustrating the temporal progression of the observations.

6 As green leaves begin to turn yellow, leaf senescence starts and is represented by the parameter DVI<sub>sen</sub>. The

7 change from green to yellow signals the transition of carbon from the leaf carbon pool to the harvest carbon pool.

8 The transition rate is simulated by reducing  $C_{leaf}$  by a specific fraction (De Vries et al., 1989),

$$C_{harv} = C_{harv} + \mu (DVI - DVI_{sen})^{\nu} \cdot C_{leaf}$$

where  $\mu$  and  $\mu$  were determined by fitting the declining trend of carbon in green leaves following leaf senescence.

11 The simulation results are presented in Section 3.1.4.

1

3

4

14

The calculation of crop height (h) depends on the amount of carbon in the stem ( $C_{stem}$ ) (Hunt, 2012):

$$h = \kappa \left(\frac{C_{stem}}{f_{C,stem}}\right)^{\lambda}$$

where  $f_{c,leaf}$  represents the carbon fraction of dry matter in the stem, and  $\kappa$  and  $\lambda$  were determined by fitting the

relationship between h and stem dry matter of stems, which is equal to  $\frac{C_{stem}}{f_{C,stem}}$  (Fig.8).

**Figure 8.** Stem dry weight against crop height. Coloured symbols are observations, and the colour shows the data from O<sub>3</sub>-FACE experiment. The black dashed line and the black solid line show the fit using parameters from Osborne et al.

(2015) and the calibrated parameters respectively. Note that various symbols correspond to successive sampling dates from

the same experimental field, thereby illustrating the temporal progression of the observations.

Similar to leaf senescence, the carbon stored in the stem reserves is mobilised into the harvest carbon pool at a rate of 10% per day, once the partition coefficient for stems drops below 0.01 (De Vries et al., 1989).

$$C_{harv} = C_{harv} + 0.1 \cdot \tau C_{stem}$$

9 where  $\tau$  represents the fraction of stem growth partitioned to reserve carbon.

The observations did not include the carbon fraction, such as  $C_{leaf}$  and  $C_{stem}$ , required for the model simulation; therefore, these values were sourced from peer-reviewed rice field studies (listed in the supplementary file). Some studies evaluated varied stressors or environmental treatments. Thus, to ensure consistency with calibration, only the control-plot values under default (unstressed) conditions were used. All the literature data were derived from rice field experiments conducted in China, involving several rice cultivars to enhance representativeness (Fig. 9). The carbon content of panicles was also obtained from literature and combined with the carbon in yellow leaves during the ripening phase to calculate the total carbon in the harvest pool. Additionally, the fractions of carbon-to-dry matter were used to compare the simulation results with the observations, which only provided dry biomass data for rice.

**Figure 9.** Carbon content of the leaf, stem, root, and panicle during different crop development stages, where the average means the value collected from the literature which only provided an average value for all stages during the rice growth. The green, yellow, and blue dashed lines represent the value prescribed in the model for the fraction of carbon to the dry matter in the root, stem, and leaf respectively.

# 3.1.4 Comparison with O<sub>3</sub>-FACE experiments

Figure 10 illustrates the changes in the main carbon pools throughout the entire growing period. The accumulated carbon was reduced under elevated O<sub>3</sub> conditions, highlighting the detrimental impact of O<sub>3</sub> on crop growth. At the maturity stage, total aboveground carbon under elevated O<sub>3</sub> was 22%–29% lower compared to ambient O<sub>3</sub> conditions, as shown in the observations (Fig. 10(e)(f)). Carbon levels in both the leaf and stem exhibited a similar decreasing trend due to the O<sub>3</sub>-induced damage to the photosynthesis process and carbon accumulation. The simulations closely matched the observations, using the average carbon-to-dry biomass fraction for different growth stages to convert observed data into carbon weights (Fig. 9). It is important to note that the carbon fraction varies with cultivar and growing environment. To align the model results, which are based on carbon weight instead of dry weight, with the observed data, the average carbon-to-dry biomass ratio across all stages was applied.

**Figure 10.** Leaf, stem, and total aboveground carbon against day of year under ambient and elevated O<sub>3</sub> conditions. Box plots are observations, whereas lines show the simulations results using parameters from Osborne et al. (2015) (grey) and calibrated (green) parameters under ambient O<sub>3</sub> conditions, including high (blue) and low (orange) O<sub>3</sub> sensitivity under elevated O<sub>3</sub> conditions, respectively, with units of g m<sup>-2</sup>.

There are two parameters in the simulations that directly relate to the impact of  $O_3$  on the rice (Clark et al., 2011; Sitch et al., 2007) (Table 5). The reduction of the net photosynthesis rate was determined by the value of the instantaneous leaf uptake of  $O_3$  above the threshold  $F_{O_3crit}$ , multiplied by a sensitivity parameter a (Pleijel et al., 2004). Observations using three planting densities of rice observations were used to calibrate the model. As can be seen from Fig.11, the high sensitivity and low sensitivities coincided with the upper and lower boundaries of relative yield (RY) which is calculated as follows:

$$RY = \frac{Y_{O_3}}{Y_0}$$

where  $Y_{O_3}$  represents the crop yield including O<sub>3</sub> damage and  $Y_0$  represents the crop yield with no effects of O<sub>3</sub>.

14 In the Fig. 11, AOT40 was used to represent the O<sub>3</sub> concentrations in the environment,

15 
$$AOT40 = \sum_{i=1}^{n} ([O_3]_i - 0.04)$$

where  $[O_3]_i$  stands for the hourly  $O_3$  concentration level (unit: ppm h) during daylight hours (08:00–19:59), and

*n* represents the total hours of the growing season.

### Table 5 O<sub>3</sub> parameters calibrated for high and low sensitivity to O<sub>3</sub> damage.

| Parameters    |                          | Osborne et | High        | Low         | The meaning of parameters                          |
|---------------|--------------------------|------------|-------------|-------------|----------------------------------------------------|
|               |                          | al. (2015) | sensitivity | sensitivity |                                                    |
| $F_{O_3crit}$ | fl_O <sub>3</sub> _ct_io | 5.0        | 7.0         | 8.0         | Critical flux of O <sub>3</sub> to vegetation      |
|               |                          |            |             |             | $(nmol m^{-2} s^{-1}).$                            |
| a             | dfp_dcuo_io              | 0.25       | 1.2         | 0.7         | Plant type specific O <sub>3</sub> sensitivity     |
|               |                          |            |             |             | parameter (nmol m <sup>-2</sup> s <sup>-1</sup> ). |

Figure 11. Relative yield against AOT40 (ppm h). Coloured lines show the relative yield of rice planted in high (green), medium (orange), and low (blue) density. The grey lines show the simulations of relative yield with high (dotted) and low (dashed) sensitivity to O<sub>3</sub> damage respectively.

Figure 12 illustrates the height and LAI of rice under both elevated and ambient O<sub>3</sub> conditions. The difference in LAI and height between these two environments underscores the negative impact of O<sub>3</sub> on rice carbon accumulation. Post-calibration, the simulations for both LAI and height align well with observational data (Fig. 11 (a)(c)). Prior to the new calibration, simulations with default parameters from Osborne et al. (2015) significantly underestimated both LAI and height largely due to the underestimated photosynthesis rate (Fig. 4). This underestimation led to reduced carbon assimilation and storage, resulting in insufficient carbon allocation to stems and leaves, which directly impacted LAI and height. It is worth noting that all plots comparing simulation and observation begin after the model's initialisation phase.

**Figure 12.** Crop height (cm) and green leaf area index (LAI) are shown versus day of year under ambient and elevated O<sub>3</sub> conditions. Box plots show observations, whereas lines show the simulations results using parameters from Osborne et al. (2015) (grey) and calibrated (green) parameters under ambient O<sub>3</sub> conditions, including high (blue) and low (orange) O<sub>3</sub> sensitivity under elevated O<sub>3</sub> conditions, respectively.

# 3.2 Evaluation

Figure 13 compares simulated and observed values of leaf carbon, stem carbon, total aboveground biomass, and rice height for the years 2022 and 2023, based on data from an independent FACE experiment (see section 2.2). The observations were limited to heading and maturity stages. These observations were compared to our newly calibrated JULES-crop model simulations using these FACE observations. The calibrated O<sub>3</sub>-damage parameters were applied to model the impact of O<sub>3</sub> on rice biomass and carbon content.

The simulated stem carbon was marginally lower than the average observed values (Fig.13 (c)(d)), while total aboveground biomass was overestimated when using low O<sub>3</sub> sensitivity parameters (Fig.13 (e)(f)). These variations can be attributed to differences in the carbon allocation between the calibration and evaluation experiments. The seeding depth notably influenced stem weight since stems thickened nearer the root, and only aboveground stems were harvested and measured. Consequently, deeper seeding resulted in a smaller fraction of stem biomass relative to total aboveground biomass (Gong et al., 2023). This slight underestimation of stem carbon was also evident in the simulation of crop height, which was similarly affected by seeding depth.

The total biomass observed in the evaluation experiment surpassed that measured in the O<sub>3</sub>-FACE experiment, particularly with a notably larger stem weight. Crop parameters were calibrated using data from the O<sub>3</sub>-FACE experiment, but differences in agronomic practices across experiments may have introduced uncertainties.

Figure 13. Leaf, stem, total aboveground biomass, and crop height against day of year for 2022 and 2023. Box plots are observations, and coloured lines show the simulation results using low (orange) and high (blue) O<sub>3</sub> sensitivity, respectively.

While the simulated crop height fell within the range of observed values, it was marginally lower than the average measured height (Fig.13 (g)(h)). Despite variations in seeding practices between the calibration and evaluation field experiments, the carbon distribution and levels aligned well with the observations. Overall, JULES-crop demonstrated the ability to accurately predict rice growth and carbon allocation across various carbon pools.

# 3.3 Limitations

While this study provides a rice model calibration based on the novel O<sub>3</sub>-FACE experiments, several limitations must be acknowledged. The calibrated O<sub>3</sub> parameters influence modelled net photosynthesis, biomass, and yield

- through the control on stomatal uptake and instantaneous photosynthesis. Due to limited O<sub>3</sub>-FACE observations,
- our calibration did not represent differences in O<sub>3</sub> sensitivity due to rice cultivar.
- Additionally, the calibrated thermal time was specific to a particular location and should be recalculated using
- local air temperature and rice phenology data if simulations are performed for other regions. For example, the
- evaluation experiment conducted in 2023 in a nearby county exhibited a relatively higher thermal time than the
- calibration experiment, primarily due to the longer growth duration. Rice growth in the 2022 and 2023 evaluation
- experiments was severely affected by crop pests and diseases at the maturity stage, leading to significant yield
- loss. As a result, only crop growth characteristics were used to validate the model.
- Furthermore, although the model was calibrated and evaluated using independent experimental data, directly
- applying the parameters to global simulations may introduce significant uncertainties. As such, global simulations
- using the parameters derived in this study should incorporate further evaluations to verify model performance
- (Müller et al., 2017).

#### 4 Conclusion

- This study marks a significant advancement in modelling rice growth and O<sub>3</sub> effects by providing the first
- calibration of the JULES-crop model using rice-specific data from O<sub>3</sub>-FACE experiments. These experiments
- offer a realistic field setting to assess the impacts of O<sub>3</sub> on crops, addressing limitations of alternative setups such
- as OTC by simulating more natural environmental conditions. Initial simulations with the default rice parameters
- in JULES-crop revealed substantial underestimation of carbon accumulation throughout the growth cycle.
- Calibration using the most recent O<sub>3</sub>-FACE data significantly improved the model's ability to replicate rice
- physiology, phenology, yield, and O<sub>3</sub> sensitivity.
- The calibration process involved adjusting key parameters to align simulations with observed data, including leaf
- area indices, crop height, yield, and the biomass of leaves, stems, and panicles. The model was refined to
- accurately represent yield reductions caused by elevated O<sub>3</sub> levels. Evaluation against independent field
- experiments demonstrated good agreement between simulated outcomes and observed results, affirming the
- model's robustness.
- This study deepens our understanding of O<sub>3</sub>'s impact on rice production and delivers a newly calibrated model
- suitable for assessing future climate scenarios and O<sub>3</sub> effects. The study lays the groundwork for future agricultural
- research aimed at mitigating O3-induced yield losses, providing a valuable framework for enhancing food security
- as O<sub>3</sub> levels continue to rise.

- Code availability. This study used the JULES (Joint UK Land Environment Simulator) version 7.4, which was
- released in November 2023. The model is available for download from the UK Met Office Science Repository
- Service (MOSRS) (https://code.metoffice.gov.uk/trac/jules), with registration required. For simulating
- photosynthesis rates, we used the Leaf Simulator (Williams et al., 2019), which is accessible at
- https://code.metoffice.gov.uk/trac/utils.

- Data availability. The calibrated driving data in this study are openly available in Zenodo at https://doi.org/
- 2 10.5281/zenodo.14008269. The O<sub>3</sub>-FACE data that supports the calibration of this study is available on request
- from the corresponding author Lianxin Yang (lxyang@yzu.edu.cn). The FACE data for evaluation is available on
- 4 request from the author Yu Jiang (yujiang@njau.edu.cn).

#### Author contributions

5

11

15

- 6 BX: methodology, formal analysis, investigation, visualisation, writing original draft preparation, revised drafts,
- 7 review & editing. SD: conceptualisation, research discussions and guidance, review & editing. HY:
- 8 conceptualisation, research discussions and guidance, review & editing. LY: provision of FACE data and advice.
- 9 YJ: provision of FACE data and advice. AC: research discussions and guidance, review & editing. KW: research
- discussions, review & editing. YW: provision of FACE data and advice. TW: provision of FACE data and advice.

## Acknowledgements

- Beiyao Xu gratefully acknowledges financial supports from the Dual Award Nanjing University/University of
- Leeds Studentship. This work used JASMIN, the UK's collaborative data analysis environment
- (https://www.jasmin.ac.uk).

# Financial support

- This work was the supported by the National Key Basic Research Development Program of China
- (2024YFC3711905). The authors are grateful for support of the National Natural Science Foundation of China
- (42477103). This work was supported by Dual Award Nanjing University/University of Leeds Studentship.

# 19 Competing interests

The authors declare that they have no conflict of interest.

22

21

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
