# Peer review of "A first calibration of the JULES-crop version 7.4 for rice"

_EGUsphere, 2024_

## Referee Comment (RC1)

This paper is well-organized and makes excellent use of the world's state-of the-art O₃-FACE experiment for rice, representing the first calibration and validation of the JULES-Crop model for rice using such data. As we know, the reliability of any process-based crop model heavily depends on experimental data that closely reflect ecological realism. Compared with controlled environment experiments, FACE experiments better replicate real-world climate conditions, including factors such as leaf temperature, humidity, and air turbulence.

This work significantly enhances the model's ability to simulate rice physiology and quantify yield losses under ozone stress, while also providing calibrated parameters that can be used by other JULES-Crop researchers, and offering a valuable reference for other crop models. It addresses a critical gap in the crop modelling literature and offers important insights for future agricultural research and climate impact assessments.

I recommend publication after the following comments are addressed:

**Major comments:**

1. P4 L34, Surface downward longwave radiation is calculated based on empirical model instead of reanalysis data, why?
2. P6 L8, Photosynthesis rate can be strongly influenced by ozone. Can you explain the reason why ozone was not considered for the calibration of photosynthesis rate?
3. P17 L13, An explanation of the role of the ozone sensitivity parameters (as listed in Table 5) could be useful. How do these parameters interpret the overall impact of ozone on rice growth?
4. P19, in Figure 13, why the height is around 40 cm, but the stem carbon is near zero?

**Minor comments:**

1. P2 L21-23, vague statement, it would be better to provide more detail on why flux-based methods demonstrate better performance than concentration-based methods.
2. P14L13 What do key symbols (e.g., κ and λ) in the equations represent and what are their units?
3. P8 L7-10, Vmax doesn't appear to be properly defined. I assume that Vmax refers to the maximum rate of Rubisco carboxylation at 25°C?
4. P10 L1-3, in the model, it is suggested that an upper temperature threshold of 38 °C begins to have a significant impact on the photosynthesis rate, as indicated by fitting the 1:1 line between simulated and observed photosynthesis rates in Figure 4. Can this upper temperature threshold be corroborated by relevant experimental literature?
5. P10 Figure 5 captions need to include brief details on data sources or experimental conditions.
6. P11 L12-13, have those values (i.e. base temperature, optimum temperature and maximum temperature) been calibrated, or are they default settings from the model?
7. P15 L3 Could you add a bit more background on the experimental fields mentioned in Figure 8?
8. P15 L5-6, again, has this leaf senescence rate been calibrated, or is it a default setting from the model?

9. P15L17 Consider providing a brief explanation of how the carbon fraction values were selected from the literature.
10. Ensure that terms like carbon pool, carbon fraction, and carbon content are used consistently.
11. Symbols in Figure 7 and Figure 8 needs captions.
12. Use °C or degree C consistently.

---

## Author Response (AR1)

**Reviewer #1:**

This paper is well-organized and makes excellent use of the world's state-of the-art O3-FACE experiment for rice, representing the first calibration and validation of the JULES-Crop model for rice using such data. As we know, the reliability of any process-based crop model heavily depends on experimental data that closely reflect ecological realism. Compared with controlled environment experiments, FACE experiments better replicate real-world climate conditions, including factors such as leaf temperature, humidity, and air turbulence.

This work significantly enhances the model's ability to simulate rice physiology and quantify yield losses under ozone stress, while also providing calibrated parameters that can be used by other JULES-Crop researchers and offering a valuable reference for other crop models. It addresses a critical gap in the crop modelling literature and offers important insights for future agricultural research and climate impact assessments.

I recommend publication after the following comments are addressed:

**Major comments:**

1. P4 L34, Surface downward longwave radiation is calculated based on empirical model instead of reanalysis data, why?

In the O3-FACE site, no direct downward longwave radiation (LW) observations were available. We therefore adopted the empirical parameterization of Chang and Zhang (2019), which was calibrated against ground-based measurements in China. This approach ensures that the LW forcing in our calibration closely reflects the local radiative environment by using observed shortwave radiation and incoming solar radiation to calculate the sky condition. We acknowledge that reanalysis products such as ERA5 provide gridded estimates of LW. It has relatively lower resolution and may underestimate the LW under all sky conditions (Chen et al., 2024), thus, we want to prioritize using the highest accuracy LW data available for the location.

2. P6 L8, Photosynthesis rate can be strongly influenced by ozone. Can you explain the reason why ozone was not considered for the calibration of photosynthesis rate?

In our four-step calibration procedure (Sect. 3.1), the first step isolates the photosynthetic parameters under non-stress conditions to avoid confounding physiological adjustments with O3 damage. By fitting the model to ambient (control) O3 measurements of photosynthesis, we ensure that fundamental biochemical parameters are tuned solely to rice physiology. The impacts of elevated O3 were then introduced via the two O3-sensitivity parameters (Table 5).

3. P17 L13, an explanation of the role of the ozone sensitivity parameters (as listed in Table 5) could be useful. How do these parameters interpret the overall impact of ozone on rice growth?

We thank the reviewer for highlighting this point. We have added a new subsection to section 2 to define and interpret these parameters in the equations and text:

JULES-crop utilises a flux-based approach to simulate the  $O_3$  damage following Sitch et al. (2007). It assumes that the potential net photosynthesis  $A_p$  was suppressed by  $O_3$ :

$$A = A_p F$$

where A is leaf-level net photosynthesis with the  $O_3$  effects and F is the reduction factor:

$$F = 1 - a \cdot max \left[ F_{O_3} - F_{O_3crit}, 0 \right]$$

where  $F_{O_3}$  represents instantaneous leaf uptake of O3.  $F_{O_3crit}$  and a are plant-functional-type-specific threshold and sensitivity factor respectively.

The O3 flux  $F_{O_3}$  (nmol m-2 s-1) is calculated as:

$$F_{O_3} = \frac{[O_3]}{r_a + [\frac{\kappa_{O_3}}{g_I}]}$$

where  $[O_3]$  (nmol m-3) is the molar  $O_3$  concentration at the reference level,  $r_a$  (s m-1) is the aerodynamic resistance and the boundary layer resistance between the leaf surface and reference level (Monin and Obukhov, 1954),  $\kappa_{O_3}$  is the ratio of leaf conductance for  $O_3$  to leaf conductance for water vapour (1.67), and  $g_l$  represents the leaf conductance for  $O_3$  as a linear function of photosynthetic rate (Cox et al., 1999):

$$g_1 = g_1^* F$$

where  $g_l^*$  is the leaf conductance in the absence of  $O_3$  effects.

**4. P19, in Figure 13, why the height is around 40 cm, but the stem carbon is near zero?**

It is because of the model's emergence-based initialisation combined with generic plant allometry, and it reflects early rice ontogeny under field conditions. Rice in JULES-crop is "born" at the emergence with only a small initial carbon pool; height then increases immediately according to the allometric rule. Observational evidence confirms that young cereal seedlings prioritize vertical elongation over biomass deposition. For example, in wheat (a close analogue to rice) plant height increased ~286 cm g-1 of stem dry weight up to 0.16 g dry weight per plant, before slowing as tillering began (Bakhshandeh et al., 2012). Rice seedlings likewise attain around 30 cm in shoot length at only 0.04 g dry weight per plant

after emergence (Liu et al., 2023). Therefore, simulated heights of ~40 cm with small stem carbon are consistent with observations of rice seedlings under field conditions.

**Minor comments:**

1. P2 L21–23, vague statement, it would be better to provide more detail on why flux-based methods demonstrate better performance than concentration-based methods.

We have expanded the introduction section by adding the following sentence:

Flux-based methods account for stomatal conductance and environmental conditions, such as temperature and vapour pressure deficit, to modify O3 uptake and thus directly link absorbed O3 dose to physiological damage. Compared with concentration-based methods, flux-based methods exhibit enhanced performance in correlating O3 levels with relative yield loss, enabling more precise assessments (Pleijel et al., 2004; Pleijel et al., 2022; Mills et al., 2011; Ronan et al., 2020).

2. P14 L13, what do key symbols (e.g.,  $\kappa$  and  $\lambda$ ) in the equations represent and what are their units?

Symbol  $\kappa$  (allo1\_io) is the allometric coefficient scaling stem-carbon content (kg m-2) to plant height (m), with units m3 kg-1, and  $\lambda$  (allo2\_io) is the unitless exponent controlling nonlinearity in that relationship. We now specify the unit of parameters in all equations throughout the paper.

3. P8 L7–10, Vmax doesn't appear to be properly defined. I assume that Vmax refers to the maximum rate of Rubisco carboxylation at 25 °C?

We have added the explanation in Sect. 3.1.1 clarifying that Vmax (mol CO2 m-2 s-1) is the maximum carboxylation rate of Rubisco, normalized to 25 °C.

4. P10 L1–3, in the model, it is suggested that an upper temperature threshold of 38 °C begins to have a significant impact on the photosynthesis rate, as indicated by fitting the 1:1 line between simulated and observed photosynthesis rates in Figure 4. Can this upper temperature threshold be corroborated by relevant experimental literature?

In JULES-crop, the upper temperature parameter ( $t_{upp}$ ) is not a hard "upper limit" on net photosynthesis but rather the temperature at which the term in the equation for the maximum rate of Rubisco carboxylation ( $V_m$ ) begins to dominate, causing modelled photosynthesis to decline sharply above this point. Parameter  $t_{upp}$  in the formulation for  $V_m$  was raised from the default 36 °C to 38 °C to achieve improved agreement between simulated and observed net photosynthesis rates in Figure 4. This adjustment is supported by experimental studies in

Figure 5 that rice photosynthesis rate increases to an optimum near 30 °C and only begins to decline sharply when leaf temperatures exceed around 38 °C. The paper has been updated accordingly.

5. P10 Figure 5 captions need to include brief details on data sources or experimental conditions.

The caption for Fig. 5 is now changed to:

The coloured lines are simulated temperature responses of photosynthesis rate using the mean value of the observed intercellular  $CO_2$  concentration of leaves ( $C_i$ ) and calibrated (38 °C) or default (36 °C)  $T_{upp}$ . The filled dots and open circles represent the observations used in this study and simulations generated by calibrated parameters, respectively. The error bars were taken from five independent studies (Supplemental Table S1), which span multiple rice cultivars, nitrogen regimes,  $CO_2$  levels, and light intensity.

6. P11 L12–13, have those values (i.e. base temperature, optimum temperature and maximum temperature) been calibrated, or are they default settings from the model?

We have clarified in Sect. 3.1.2 that the base (8 °C), optimum temperature (30 °C), and maximum (42 °C) temperatures remain the values from Osborne et al. (2015).

7. P15 L3 Could you add a bit more background on the experimental fields mentioned in Figure 8?

In the revised manuscript, we now note that observations from Figure 8 derive from O3-FACE experiment, which employed different planting density and O3 concentrations.

8. P15 L5–6, again, has this leaf senescence rate been calibrated, or is it a default setting from the model?

We confirm that the leaf senescence allometric coefficients were calibrated using the trend of green-leaf carbon decline in our FACE data.

9. P15 L17 Consider providing a brief explanation of how the carbon fraction values were selected from the literature.

A sentence has been added in Sect. 3.1.3:

Carbon-to-dry-matter fractions for leaves, stems, roots and panicles were obtained by compiling measurements from peer-reviewed rice field studies—many of which evaluated varied stressors or environmental treatments—and, to ensure consistency with our calibration,

only the control-plot values under default (unstressed) conditions were used; the full list of sources is provided in the Supplementary file.

10. Ensure that terms like carbon pool, carbon fraction, and carbon content are used consistently.

We have performed a thorough editorial pass to harmonize terminology. "Carbon pool" refers exclusively to modelled state variables, "carbon fraction" to literature-derived carbon-to-dry-matter ratios (unit: kg C kg-1 dry weight), and "carbon content" is changed to "carbon fraction". All occurrences have been standardized.

**11. Symbols in Figure 7 and Figure 8 need captions.**

We have updated the captions of Figures 7 and 8 to note that the various symbols correspond to successive sampling dates from the same experimental field, thereby illustrating the temporal progression of the observations.

**12. Use °C or degree C consistently.**

We have standardised all temperature units to °C throughout the manuscript, including figures, tables, and text.

**Reviewer #2:**

**General Comments:**

This manuscript presents a novel calibration of the JULES-crop model for rice, incorporating data from a new Free Air Concentration Enrichment (FACE) experiment with elevated ozone (O3). The use of field-based O3-FACE data to tune crop-model parameters is timely and important, given increasing O3 stress on agriculture. The authors outline clear objectives and proceed through logical calibration steps (leaf-photosynthesis, phenology, carbon partitioning, growth, and O3 response), followed by an evaluation on independent field data. In general, the conclusions—that calibrated parameters improve simulated crop growth and O3 impacts relative to defaults—are supported by the results. The writing is clear, and the figures convey the key points, but there are some issues that should be addressed.

The authors leverage unique O3-FACE observations, which is a major strength. The calibration includes tuning of two O3-response parameters using three planting densities. It would be useful to explain how "high" and "low" sensitivity bounds were chosen (e.g. to bracket the observed RY range) and whether a single "best" set was identified.

We thank the reviewer for this important suggestion. In the revised manuscript (Section 3.1.4), we have clarified that the high and low ozone sensitivity parameter sets were chosen to bracket the full range of observed relative yield (RY) reductions from the three planting densities. A single 'best' parameter set was not selected, as the planting density had a strong influence on the observed RY, and the range-based approach better captures the uncertainty in field response.

In the Abstract and Introduction, the authors claim to calibrate "O3-response parameters", but the manuscript might clarify that only those two parameters were tuned.

It is true that two key O3-response parameters in JULES-crop were calibrated, but these exert a strong, season-long influence on carbon assimilation and crop growth. In the JULES-crop flux-based O3 scheme, any stomatal O3 uptake above threshold reduces instantaneous net photosynthesis and stomatal conductance. Thus, calibrating two key O3-response parameters effectively changes both O3 uptake and the carbon assimilation reduced by O3 at each timestep and modifies the crop response at the next time step. In practice, calibrating these two parameters substantially altered simulated leaf carbon assimilation, canopy development and final biomass (Figs. 11–13). In short, even a modest change in two key O3-response parameters can influence crop growth and yield throughout the life cycle.

More discussion of uncertainties in these parameters and how they affect model outputs would be valuable.

The calibrated O3 parameters influence modelled net photosynthesis, biomass, and yield through the control on stomatal uptake and instantaneous photosynthesis. Due to limited O3-FACE observations, our calibration did not represent differences in O3 sensitivity due to rice cultivar. For example, hybrid cultivars showed greater yield loss due to O3 than inbred cultivars (Shi et al., 2009; Feng et al., 2022). Hence, modelled O3 impacts should be interpreted with caution and, where possible, constrain parameters with additional physiological data or experiments to reduce this uncertainty. The paper has been updated accordingly.

Overall, I find the work to be worthy of publication after revision. The study tackles an important problem (modelling O3 effects on rice) and makes a valuable contribution. My detailed, section-specific suggestions below should help improve clarity, completeness, and rigour.

**Specific Comments:**

1 In introduction, the motivation is well explained. One phrase could be improved: "Rice is the primary energy source for over half of the world's population...". Rice is typically called a staple food or calorie source rather than "energy source."

**Corrected. The sentence now reads:**

Rice is the staple food for over half of the world's population and plays a crucial role in global food security.

2 Section 2.1, since JULES-crop's O3 damage scheme is central, I suggest briefly summarizing it here or adding a forward reference to Sect.3.1.4. For readers not familiar with JULES-crop, clarify how O3 effects are implemented (e.g. via reduced assimilation tied to stomatal O3 flux).

We thank the reviewer for highlighting this point. We have added a new subsection to section 2 to explain the O3 damage scheme in JULES-crop.

3 In Section 2.2, You might add the actual O3 concentration levels (ambient vs elevated) to quantify "25% higher."

The mean daytime O3 concentration during the experimental period was approximately 46 ppb under the elevated O3 treatment, compared to 37 ppb in the ambient environment—an increase of around 25%. The paper has been updated accordingly.

4 The default values of ratio of stem/root nitrogen concentration to leaf nitrogen concentration and calibrated values indicate leaf has more N than stem/root – is this consistent with literature?

Thank you for raising this point. For rice, leaf tissues generally contain higher nitrogen concentrations than stems or roots except at senescence when leaves yellow and remobilize N. In our calibration, we derived the stem-to-leaf and root-to-leaf ratios from measurements taken before maturity, deliberately excluding the senescing-leaf stage. By doing so we focused on the period when leaves remain green and actively photosynthesizing, which is the phase these parameters govern in JULES-crop. Literature on rice N partitioning supports this approach (De Vries et al., 1989). Thus, our calibrated ratios align well with published values for active growth stages and ensure accurate modelling of photosynthetic capacity.

5 The figures are relevant, but some captions and labelling need improvement. For example, Figure 12's caption contains a typo ("verses day of year" should be "versus"). Several figure captions should clearly define all symbols, line colours, and panels. The axes and units should be legible. In Fig.12 and 13, the color-coding is explained, but ensure consistency (e.g. in Fig.13 caption "low" and "high" O3 sensitivity should specify which line is which colour).

Corrected. All figure captions have been reviewed and revised.

All symbols and colours are now explicitly defined in each caption.

6 Spelling is inconsistent (e.g. "ozone" vs "O3").

We have standardized terminology throughout the text and figures.

7 Minor grammatical fixes (e.g. "As green leaves begin to turn yellow..." instead of "As green begin to turn yellow") would improve readability.

Corrected.

8 Ensure all acronyms (LAI, FACE, POD, AOT40, etc.) are defined at first use.

All acronyms are now defined on first use, including:

LAI: Leaf Area Index

FACE: Free Air Concentration Enrichment

POD: Phytotoxic Ozone Dose

AOT40: Accumulated exposure over a threshold of 40 ppb

9 The section heading "Result" should be plural (Results).

Corrected. The section is now titled "Results".

10 In Sect.3.1.4, the term "O3 related parameters were applied" (p.18) could be rephrased more clearly (e.g. "The calibrated O3-damage parameters were applied").

Corrected.

In summary, I find this manuscript to be a valuable contribution on calibrating a crop model with novel O3 data. Addressing the above points will significantly enhance the paper. After revision, it should be suitable for publication in Geoscientific Model Development.

We again thank the reviewer for the constructive comments. We believe the revisions have significantly improved the clarity, rigor, and completeness of the manuscript.

**References**

- Bakhshandeh, E., Soltani, A., Zeinali, E., and Kallate-Arabi, M.: Prediction of plant height by allometric relationships in field-grown wheat, Cereal Research Communications, 40, 413-422, 2012.
- Chen, Y., Jiang, B., Peng, J., Yin, X., and Zhao, Y.: Evaluation of the Surface Downward Longwave Radiation Estimation Models over Land Surface, Remote Sens., 16, 3422, 2024. Cox, P., Betts, R., Bunton, C., Essery, R., Rowntree, P., and Smith, J.: The impact of new land surface physics on the GCM simulation of climate and climate sensitivity, ClDy, 15, 183-203, 10.1007/s003820050276, 1999.
- De Vries, F. P., Jansen, D., Ten Berge, H., and Bakema, A.: Simulation of ecophysiological processes of growth in several annual crops, Int. Rice Res. Inst. 1989.
- Feng, Z., Xu, Y., Kobayashi, K., Dai, L., Zhang, T., Agathokleous, E., Calatayud, V., Paoletti, E., Mukherjee, A., Agrawal, M., Park, R. J., Oak, Y. J., and Yue, X.: Ozone pollution threatens the production of major staple crops in East Asia, Nat. Food, 3, 47-56, 10.1038/s43016-021-00422-6, 2022.
- Liu, Y., Yao, Y., Yang, Y., Shi, G., Ding, F., Liu, G., Zhang, S., Xie, J., Yu, Z., and Li, S.: Small molecular organic acid potassium promotes rice (Oryza sativa L.) photosynthesis by regulating CBC and TCA cycle, Plant Growth Regulation, 101, 569-584, 10.1007/s10725-023-01041-w, 2023.
- Mills, G., Hayes, F., Simpson, D., Emberson, L., Norris, D., Harmens, H., and Bueker, P.: Evidence of widespread effects of ozone on crops and (semi-)natural vegetation in Europe (1990–2006) in relation to AOT40- and flux-based risk maps, Glob. Chang. Biol., 17, 592-613, <a href="https://doi.org/10.1111/j.1365-2486.2010.02217.x">https://doi.org/10.1111/j.1365-2486.2010.02217.x</a>, 2011.
- Monin, A. and Obukhov, A.: Osnovnye zakonomernosti turbulentnogo peremeshivanija v prizemnom sloe atmosfery (Basic laws of turbulent mixing in the atmosphere near the ground), Trudy geofiz. inst. AN SSSR, 24, 163-187, 1954.
- Pleijel, H., Danielsson, H., and Broberg, M. C.: Benefits of the Phytotoxic Ozone Dose (POD) index in dose-response functions for wheat yield loss, Atmos. Environ., 268, 118797, https://doi.org/10.1016/j.atmosenv.2021.118797, 2022.
- Pleijel, H., Danielsson, H., Ojanperä, K., De Temmerman, L., Högy, P., Badiani, M., and Karlsson, P.: Relationships between ozone exposure and yield loss in European wheat and potato—a comparison of concentration-and flux-based exposure indices, Atmos. Environ., 38, 2259-2269, 2004.
- Ronan, A. C., Ducker, J. A., Schnell, J. L., and Holmes, C. D.: Have improvements in ozone air quality reduced ozone uptake into plants?, Elementa: Sci. Anthropocene, 8, 10.1525/elementa.399, 2020.
- Shi, G., Yang, L., Wang, Y., Kobayashi, K., Zhu, J., Tang, H., Pan, S., Chen, T., Liu, G., and Wang, Y.: Impact of elevated ozone concentration on yield of four Chinese rice cultivars under fully open-air field conditions, Agriculture, Ecosystems & Environment, 131, 178-184, <a href="https://doi.org/10.1016/j.agee.2009.01.009">https://doi.org/10.1016/j.agee.2009.01.009</a>, 2009.
- Sitch, S., Cox, P. M., Collins, W. J., and Huntingford, C.: Indirect radiative forcing of climate change through ozone effects on the land-carbon sink, Nature, 448, 791-U794, 10.1038/nature06059, 2007.